

# PoP
## Project of Projects

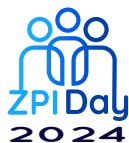

**Autors**: Zuzanna Aszkiełowicz ⬤ · Nouran Elmenshawy ⬤ · Yimeng Liu ⬤ · Weronika Wójcik⬤

**Supervisor:** Krystian Wojtkiewicz ⬤

### Abstract

The project is titled **Project of Projects** (acronym PoP), as it is a project designed to serve as a management tool for the assessment of Team Projects (Zespołowe Przedsięwzięcie Inżynierskie—ZPI), which is undoubtedly a critical aspect of engineering education at the Wroclaw University of Science and Technology. The platform addresses inconveniences of current practices (such as file uploading and transparency in feedback) by automating workflows and enhancing collaboration among students, supervisors, reviewers, and event chairs. Key features include role-based access, project management workflows, weighted evaluations, edition-based filtering, and a comprehensive statistics dashboard.

The platform improves efficiency, saves time, and ensures project evaluation and feedback transparency. With modern technologies, PoP delivers a purpose-built solution. Enhancing organization and communication ensures a collaborative environment for a more effective and transparent ZPI management process. Future improvements include a mobile app, a Polish version, and offline access.

## 1   INTRODUCTION

The Team Project is a pillar of engineering education, encouraging cooperation among stakeholders–students, supervisors, reviewers, event chairs, and spectators. Despite its importance, the existing process of organizing, managing and evaluating ZPI projects often involves significant manual effort and inefficiency. These difficulties include disjointed communication, time-consuming project management, and a lack of transparency in feedback and assessment.

To address these issues, our team developed an integrated platform that would simplify ZPI management and organization. This system focuses on essential activities that allow students to form teams, submit project files, and receive feedback. Supervisors and reviewers will benefit from automated workflows for project evaluation, while event chairs and spectators will have easy access to view relevant information about the team projects.

The platform uses JWT [9] for state-free secure authentication and authorization. Additionally, the integration of OAuth2 [7] with Google and OAuth1.0 [2] with USOS [3] ensures access for different types of user, e.g., spectators (the former) and university-related users (the latter, through university credentials). We used Granted Authority [1] to secure API call access.

The primary objectives of our projects are to:

**Improve Efficiency** through task automatization such as project submission, invitation management, and enhanced stakeholder collaboration to reduce administrative burden.

**Save Time** Organize all project-related information in a single platform to minimize the time spent searching and managing files.

**Enhance Collaboration** by providing better interaction with features such as messaging and real-time project status updates.

**Provide Transparency** , i.e., ensure stakeholders have straightforward and easy access to view relevant information about the team projects, so everything is handled openly and fairly, reducing biases and misunderstandings.

## 2   RELATED WORK

### 2.1   Existing Solutions and Technologies

When developing PoP, we began by exploring existing tools that might address similar challenges. While platforms such as EasyChair [4], Google Scholar [5], and ResearchGate [12] offer valuable functionalities in their respective domains, none adequately meet the specific needs of our faculty, particularly

in managing student projects. Let's consider EasyChair which is widely used for managing academic conferences, offering features like paper submissions, peer reviews, and feedback coordination. However, its design is tailored to conference workflows, making it unsuitable for the detailed and iterative processes involved in supervising and evaluating student projects.

Google Scholar and ResearchGate serve as essential platforms for researchers to share their work and build professional profiles. However, they lack the tools necessary for submission management, role-based workflows, and feedback coordination, which are crucial in an educational context.

PoP was built from the ground up to address these gaps, focusing exclusively on our faculty's needs. Designed with input from all stakeholders—students, supervisors, reviewers, and chairs—it integrates all necessary functionalities into a single platform. Unlike general-purpose academic tools, POP is tailored to manage student projects comprehensively, from submissions and evaluations to feedback and role assignments.

Key features of PoP include role-based permissions that ensure users—such as students, supervisors, and reviewers—access only the information relevant to their roles. The system also supports structured workflows for evaluations and detailed feedback mechanisms, enabling transparency and efficiency. While platforms like EasyChair or ResearchGate excel in their domains, POP's specialized design provides the targeted solutions our faculty requires for managing team projects.

## 2.2 Technology Choices

Technologies have been thoroughly researched and selected based on their technological advancement and potential. They will be presented according to specific application layers.

### Database

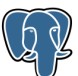

**PostgreSQL** [6] In our project, we identified several entities, such as *users, projects, edits, evaluations, invitations*, etc., along with relationships among them. Therefore, we needed a relational database management system that could sufficiently deal with complex queries, joins, and foreign key constraints while ensuring data integrity – PostgreSQL meets all these requirements.

### Backend

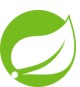

**Spring Boot** [14] Spring Boot was our choice not only because of our Java knowledge but also because of the features it offers. We wanted to create a secure application, and Spring Boot makes it easy to implement authentication using OAuth2 and OAuth 1.0 and authorization management. Additionally, we appreciated its excellent integration with various data sources, including PostgreSQL. Thanks to the "convention over configuration approach", we could focus on the simple configuration of RESTful APIs, which accelerated our work on the project.

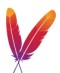

**Maven** [11] To improve dependency management, we chose Maven, which enables efficient and seamless integration of various libraries and frameworks–in our case, Spring Boot, MyBatis (for database communication), PostgreSQL, and many others. Maven allowed us to easily manage dependencies such as OAuth2 client support, security configurations, database integration, and overall project consistency. Additionally, using the Spring Boot Maven plugin simplifies the application packaging and deployment process.

### Frontend

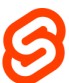

**Svelte** [13] While new to most team members, Svelte quickly proved to be an excellent choice for the front end [10]. Its lightweight nature and intuitive approach to reactivity enabled us to build a dynamic and responsive user interface with ease.

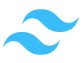

**Tailwind CSS** [8] Tailwind CSS streamlined the design process by providing a comprehensive set of utility classes. This eliminated the need for writing custom CSS from scratch, ensuring a cohesive and consistent visual design while saving development time.

## 2.3 Time Constraints

**Weeks 1–2** Project setup, repository configuration, and defining technical requirements.

**Weeks 3–5** Core functionalities like authentication, invitation flow, and project upload.

**Weeks 7–8** Advanced features such as evaluation systems and search functionalities.

**Final Weeks** Testing, debugging, and preparing documentation.

## 2.4  Resources

**Human Resources** – Our team had four members, each with their own focus. We divided responsibilities among frontend, backend, and database development (Tab. 1).

**Technical Resources** – We relied on Git for version control and collaboration, and our stack—Svelte, Tailwind CSS, Spring Boot, and PostgreSQL—handled the development side.

| Team Member | Role and Contributions |
|---|---|
| Yimeng Liu | Database, backend |
| Weronika Wójcik | Backend |
| Zuzanna Aszkiełowicz | Backend |
| Nouran Elmenshawy | Frontend |

Table 1: Project Team and Their Contributions

# 3  RESULTS

The development of the PoP application has been successfully completed, resulting in a tailored solution designed specifically to address the unique needs of our faculty in managing student projects. The platform integrates various functionalities to streamline workflows, enhance collaboration, and provide comprehensive tools for submissions, evaluations, and feedback.

In the upcoming subsections, we will delve into the key components of the PoP system, highlighting its core features, architecture, and the design decisions that make it uniquely suited to our requirements. Each component will be discussed in detail, providing insights into how PoP facilitates efficient project management and improves the overall user experience for all stakeholders involved.

## 3.1  Role-Based Functionalities and Data Access

| Role | Key Features | Access Level |
|---|---|---|
| Student | Submit project, receive feedback | Limit to their team |
| Supervisor | Evaluate projects, manage team members | Full project access |
| Chair | Assigned supervisors and reviewers | Admin Access |
| Reviewer | Provide Evaluations | Full project access |
| Spectator | Browse projects | Limited view and evaluation |

Table 2: Roles and their access levels

It is worth noting that role settings have significant flexibility and scalability in our project, primarily reflected in:

**Edition-role concept** – User roles can change according to edition changes.

**Concept of multiple roles** – The same user can have one or multiple roles, aligning the application with actual needs.

## 3.2  Project Management

In PoP's project management process, we use a series of workflows to ensure each project has relevant personnel involved at different stages. The specific process is as follows:

- The Chair creates a project and sends an invitation to the relevant supervisor.

- The Supervisor accepts the invitation, joins the project and invites the students who are involved in the project development to join the project.

- Students accept the invitation, join the project, fill in the basic information of the project and upload the attachments required by the project.

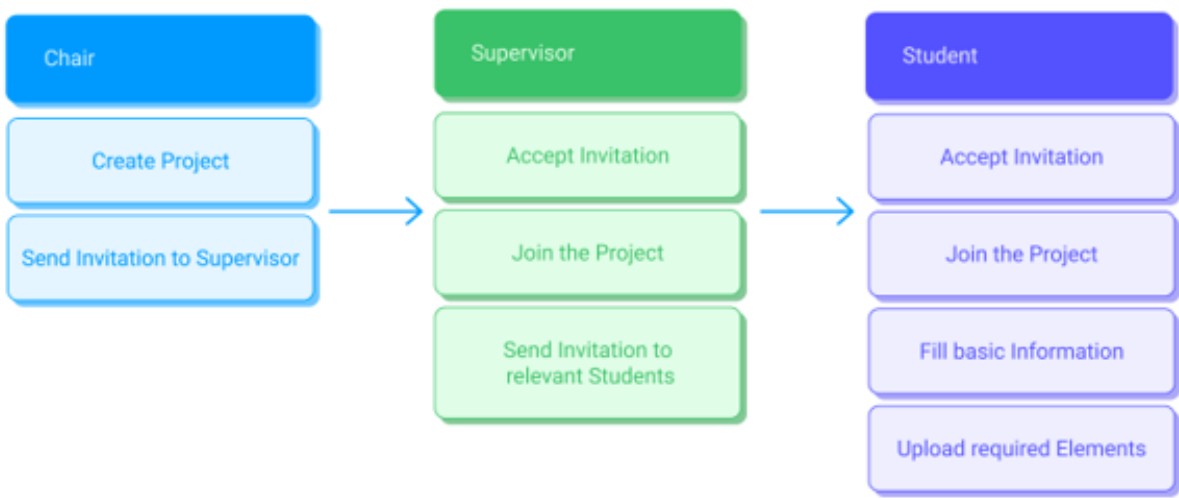

Figure 1: Workflow presentation

### 3.2.1 Evaluation and Review

During the project review and evaluation phase, we are committed to the design concept of "openness and professionalism" coexisting, and also allow specific evaluation weights to be defined according to different roles to ensure the scientificity and objectivity of the evaluation results.

- In terms of openness, the system allows all users to participate in comments and ratings within their authority to encourage extensive communication and interaction.

- In terms of professionalism, we ensure that key roles (such as supervisors and reviewers) play a leading role in the evaluation.

Following the technical implementation of the **Dynamic weight adjustment based on roles stored in the** `evaluation_weights` configuration is presented with an example of `evaluation_weights` configuration:

| Role | Weight |
|---|---|
| evaluation.weights.ASSIGNED_TO_EVALUATE | 0.35 |
| evaluation.weights.SUPERVISOR | 0.35 |
| evaluation.weights.GENARAL_TEACHING_MEMBER | 0.2 |
| evaluation.weights.STUDENT | 0.075 |
| evaluation.weights.SPECTATOR | 0.025 |

Table 3: Example of Role-Based Evaluation Weights Configutation

### 3.2.2 Edition-Based Filtering

In order to adapt to version changes, we have added edition to many concepts such as "project, role, deadline". It makes possible for the system to clearly record and distinguish different editions of project data during multi-semester or multi-year use, thereby supporting more flexible management and expansion.

- In terms of "project", edition is marked with each project, ensure that the project's contents accurately correspond to a specific time.

- In terms of "role", the concept of version allows the same user to play different roles in different semesters or project editions.

- In terms of "deadline", based on edition, each element type can set independent soft and hard deadlines according to different semesters or years.

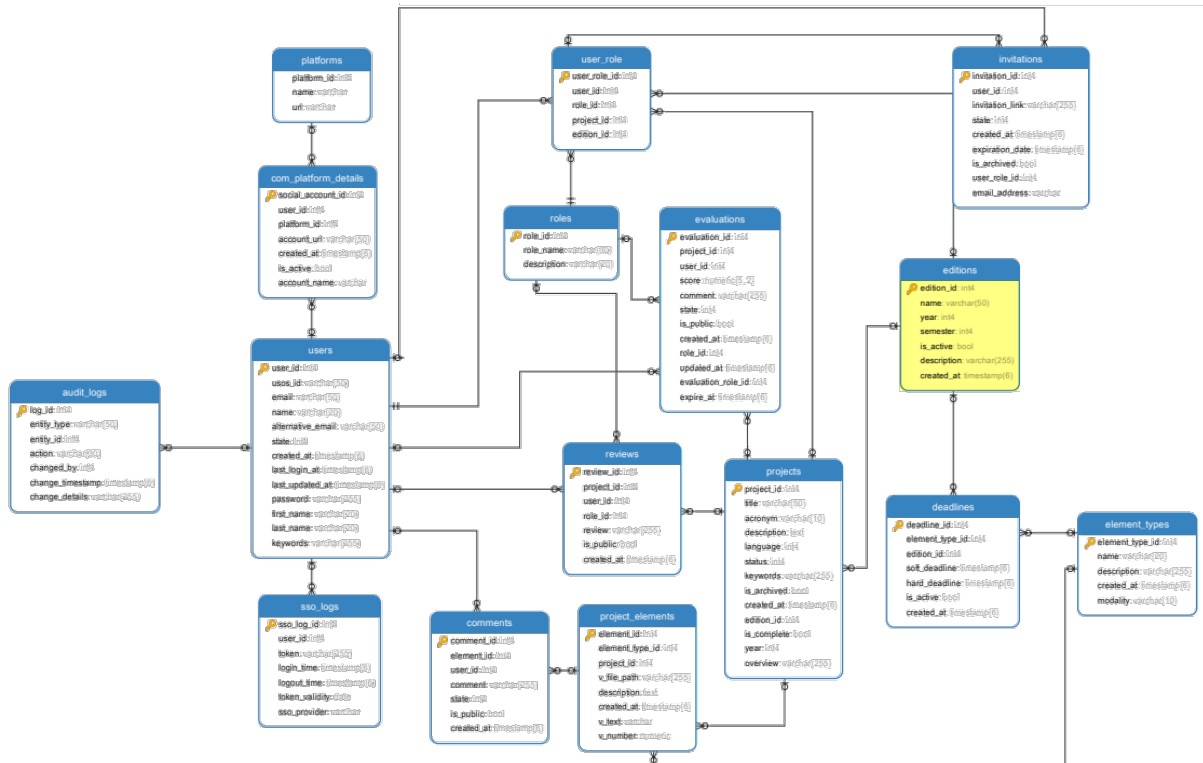

Figure 2: Database design related to Editions

Below, we outline the specific functionalities and interfaces enabled by this design, referencing the provided API endpoints for clarity:

| Title | API Endpoint | Key Implementation (Tables Involved) |
|---|---|---|
| Edition-Based Project Filtering | GET /project/getByUserRole | projects, editions, user_role |
| Edition-Specific Deadlines | GET /deadlines/getDeadlineByProjectIdAndElementTypeId | deadlines, element_types, projects, editions |
| Edition Management | POST /editions/add | editions |
| Edition-Based Statistics | GET /statistic/getCounts
GET /statistic/averageGrades | projects, editions, evaluations, roles |
| Edition-Specific Role Management | POST /userRole/removeStudentsFromProject | user_role, projects, editions |
| Edition-Based Element Management | POST /projectElements/uploadElement
GET /projectElements/retrieve | project_elements, element_types, projects, editions |
| Edition-Specific Evaluations | POST /evaluations/add
GET /evaluations/assignedEvaluateList | evaluations, projects, editions, user_role |
| Edition-Specific Comments | POST /comments/add
GET /comments/getByElementId | comments, project_elements, projects, editions |

Table 4: API Endpoints and Key Implementations

### 3.2.3 Statistics Dashboard

To provide users of various roles with a more intuitive visual presentation, we introduced the "Dashboard" page to show users what they may care about most.

The following features are introduced to the dashboard page:

- **Dynamic filters**: Users can customize the data display range based on semester, project type, role, etc.

- **Data visualization**: Intuitively present key indicators and trends through various chart forms.

- **Quick entry for operation**: Users can jump directly from the dashboard to a specific operation page, e.g., "view my teams" or "view evaluations for all projects".

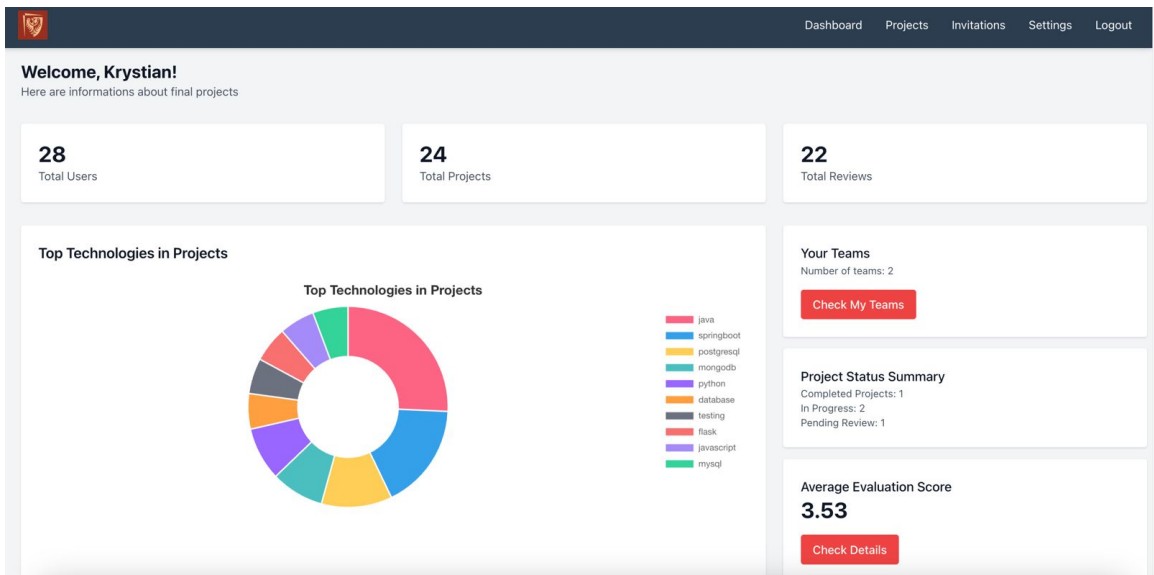

Figure 3: Dashboard View

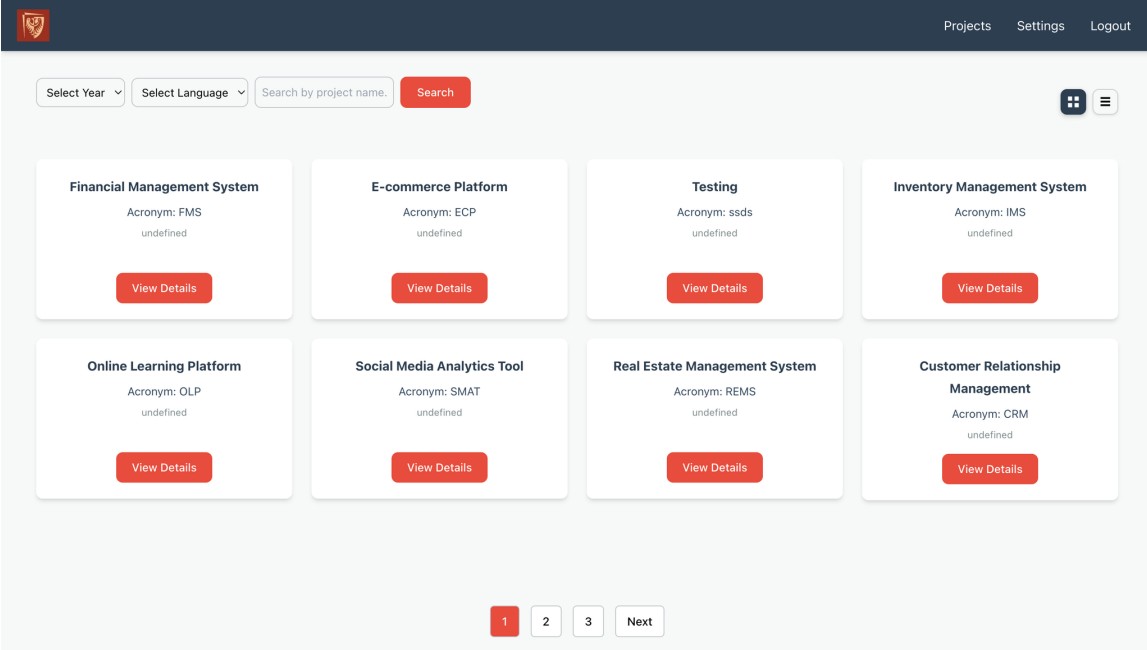

Figure 4: Projects View

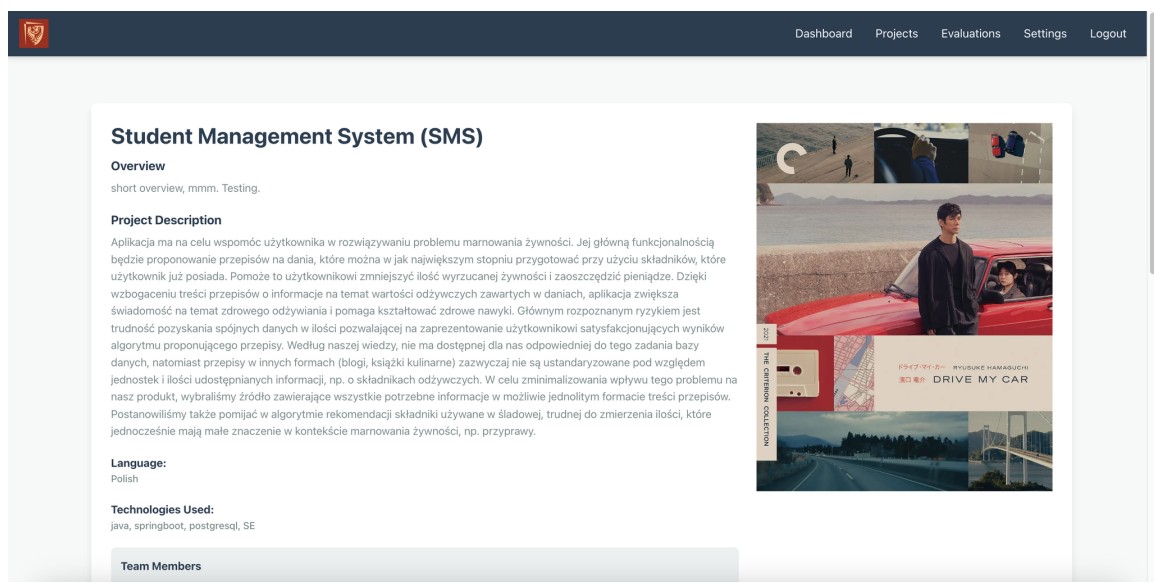

Figure 5: Project Details View

# 4  CONCLUSION

The Project of Projects system addresses the challenges of managing student Team Projects in an academic environment by providing a platform that enhances organization, management, and evaluation. Explicitly designed for ZPI, the system supports collaboration between students, supervisors, reviewers, and chairs through various features that improve efficiency and coordination. It offers flexible and scalable role management, allowing for dynamic roles based on project editions and multiple-role user assignments. The project management workflow is streamlined, with transparent processes for project creation, role assignments, and collaboration.

The evaluation and review process is structured to ensure scientific and objective assessments, with weighted evaluations tailored to different roles. The system encourages interaction among users while maintaining professionalism to guarantee reliable results. PoP also allows seamless project data management across multiple semesters or academic years, making it adaptable and organized. A statistics dashboard provides visualized data trends, dynamic filtering, and quick access to key operations, ensuring that users are informed and empowered.

From a business perspective, the system reduces administrative overhead and improves transparency and collaboration, enabling faculty and staff to focus more on academic value rather than logistical tasks. From a technical perspective, PoP highlights the integration of advanced technologies such as Spring Boot, PostgreSQL, Svelte, and Tailwind CSS, resulting in a solid, maintainable, and scalable solution.

## 4.1  Future Directions

**Mobile App**   It might be a good idea to eventually add a mobile app. This could make it easier for students and supervisors to handle things like submissions and feedback from their phones, especially when they're not at their computers.

**Polish Version**   We've thought about making a Polish version of the system. It could help local users who are more comfortable working in Polish and make things simpler for them.

**Offline Access**   Adding offline access could be useful too. It would let people check details, submit files, or review things even without internet. Once they're back online, everything could sync up automatically. The project was developed under the supervision of Krystian Wojtkiewicz, Ph.D.

# ACKNOWLEDGMENTS

We extend our heartfelt gratitude to our supervisor, Krystian Wojtkiewicz, Ph.D., for his invaluable guidance and insightful suggestions throughout the development of the Project of Projects. His support was instrumental in helping us elevate the project to its current level of excellence.

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
