# OpenReview forum: "Project of projects"
_pwr.edu.pl/Wrocław_University_of_Science_and_Technology/2024/ZPI_Day — Wrocław University of Science and Technology 2024 ZPI Day Submission_

### Official Review · Reviewer_UCHD · 2024-12-05
**Project of projects**

**Confidence:** 4
**Significance Of Results:** 5
**Overall Quality:** 4

**Compliance With Template:**

5: Very High Quality – The article contains all the required sections, which are written in a very detailed, clear, and error-free manner. The structure is professional and meets expectations, and the content adheres to the highest substantive and formal standards.

**Description Of Results:**

4: High Quality – The results are described in detail and supported by usage examples or evaluations. The description is reliable but may lack full depth of analysis.

**Feedback On Consistency:**

The project description for the Project of Projects (PoP) platform is clear and logically organized. It starts with a well-defined problem statement, explaining how current methods for managing Team Projects aka ZPI at Wroclaw University of Science and Technology are inefficient due to manual workflows and lack of transparency. The objectives are clearly outlined, aiming to automate workflows, reduce administrative burdens, and improve collaboration among students, supervisors, and reviewers.

The presentation of results aligns well with these objectives. The key features, such as role-based management, weighted evaluations, edition-based filtering, and a statistics dashboard, are presented as effective solutions to the identified problems. The conclusions summarize how PoP addresses these challenges, highlighting improvements in efficiency, transparency, and collaboration.

However, there are some areas where the consistency could be enhanced. While the article mentions gaps and limitations, such as the lack of empirical validation and missing features like mobile support, these issues are not fully addressed in the conclusions. Additionally, the comparison with existing solutions like EasyChair could be more detailed to emphasize PoP’s advantages.

Overall, the problem analysis, results, and conclusions are mostly consistent and logical. Strengthening the links between the identified problems and how the features of PoP effectively solve them would make the project description even more cohesive.

NOTE: In this article, some statements are not needed, in my opinion, like repeating the fact "The project was developed under the supervision of Krystian Wojtkiewicz, Ph.D." at the end of  Future Directions, which is clearly stated at the beginning of the article.

**Potential For Development:**

The article indicates several promising possibilities for further development and practical applications:
- Mobile Integration: Developing a mobile application would enhance accessibility, allowing users to engage with the platform on the go. This is especially important for students and supervisors who rely heavily on smartphones.
- Polish Language Support: Adding support for the Polish language would make the platform more user-friendly for local stakeholders, potentially increasing adoption rates within the university.
- Offline Access: Implementing offline functionalities would allow users to access and interact with the platform even without a stable internet connection, improving reliability and user experience.

From my side, I would strongly recommend a better and more detailed competitor analysis and event storming with a broader audience and proper stakeholders (including students, ZPI group supervisors, seminars lecturers, coordinators, and reviewers.

By focusing on these areas, the PoP platform has the potential to become a more robust and widely adopted solution for team project management. These developments would not only enhance its functionality but also broaden its applicability, making it valuable to a larger audience.

**Project Nature Evaluation:**

The PoP platform presents key characteristics of engineering work. It provides a practical solution to real-world problems faced in managing team projects at a university. The level of utility is high, as it aims to streamline workflows, reduce administrative tasks, and enhance collaboration among various stakeholders.

Unfortunately, I have the impression that the Authors of the paper did not fully understand the application requirements, which were most likely imposed by their supervisor. I would also suggest analyzing solutions like https://sessionize.com/ for handling commercial conferences and CFP management.

Technically, the project applies advanced methods and technologies. The backend uses Spring Boot with secure authentication protocols like OAuth2 and JWT, ensuring data security and integrity. PostgreSQL is used for managing relational data, which is suitable for handling complex relationships between users, roles, projects, and evaluations. The frontend is developed with Svelte and Tailwind CSS, offering a responsive and efficient user interface.

Role-based access control, weighted evaluations, and dynamic filtering demonstrate thoughtful application of engineering principles to solve specific problems. By integrating these technological solutions, the project shows a strong understanding of both the technical and practical aspects needed to develop a functional and scalable system.

**Technical Language Precision:**

4: High Quality – The language is appropriate for a technical report. Terminology is used correctly, and statements are precise, with only minor shortcomings that do not affect the overall clarity.

---

### Official Review · Reviewer_LeoA · 2024-12-05
**Judgement for projects is coming**

**Confidence:** 5
**Significance Of Results:** 5
**Overall Quality:** 4

**Compliance With Template:**

5: Very High Quality – The article contains all the required sections, which are written in a very detailed, clear, and error-free manner. The structure is professional and meets expectations, and the content adheres to the highest substantive and formal standards.

**Description Of Results:**

4: High Quality – The results are described in detail and supported by usage examples or evaluations. The description is reliable but may lack full depth of analysis.

**Feedback On Consistency:**

The paper identifies the goals and results. They are laid out and discussed clearly and precisely.  I would keep the information regarding the supervisor within the limit.

**Potential For Development:**

Hopefully, the project will be further developed, and next year, ZPI-Day will be utilizing it.

**Project Nature Evaluation:**

The project is a good example of engineering work in terms of properly identifying and comforting stakeholders' needs. However, certain issues should be addressed:
1. The quality of images could be better
2. The identification of objectives could be more stakeholder-oriented. Now, it seems to be a bit generic.
3. The choice of related solutions seems awkward. Google Scholar and ResearchGate, are not really in the same area.
4. It would be great if the authors could indicate if the project has already been deployed.

**Technical Language Precision:**

5: Very High Quality – The language is entirely appropriate for a technical report. All terms are used correctly and precisely, and the style is professional, clear, and coherent, without any errors or ambiguities.

---

### Official Review · Reviewer_pB2t · 2024-12-06
**Projekt przedstawia ciekawą platformę do zarządzania projektami studentów. Zaprezentowane rozwiązanie łączy wiele interesów i jest niezwykle użyteczne z punktu widzenia liczby przygotowywanych projektów.**

**Confidence:** 2
**Significance Of Results:** 4
**Overall Quality:** 5

**Compliance With Template:**

5: Very High Quality – The article contains all the required sections, which are written in a very detailed, clear, and error-free manner. The structure is professional and meets expectations, and the content adheres to the highest substantive and formal standards.

**Description Of Results:**

5: Very High Quality – The results are described in detail, clearly and comprehensively, supported by thorough evaluation, analysis, and convincing usage examples. The description meets the highest substantive standards.

**Feedback On Consistency:**

W przedstawionym projekcie zidentyfikowano lukę, którą zaspokaja przedstawione rozwiązanie. Zaprezentowane efekty są użytkowe oraz zaspokajają potrzeby wielu stron zaangażowanych w proces.

**Potential For Development:**

Wiele rezultatów zostało zaimplementowanych i sprawdzonych. Warto rozważyć wykorzystanie aplikacji w szerszym zakresie, jednocześnie upraszczając procesy rejestracji poszczególnych osób z procesu.

**Project Nature Evaluation:**

Przygotowane rozwiązanie wykorzystuje szeroki wachlarz rozwiązań technicznych, co potwierdza jego akceptację jako pracy inżynierskiej.

**Technical Language Precision:**

5: Very High Quality – The language is entirely appropriate for a technical report. All terms are used correctly and precisely, and the style is professional, clear, and coherent, without any errors or ambiguities.

---

### Official Review · Reviewer_qGhZ · 2024-12-06
**The described project has a chance to be useful when conducting subsequent editions of the course.**

**Confidence:** 3
**Significance Of Results:** 4
**Overall Quality:** 4

**Compliance With Template:**

4: High Quality – The article contains all the required sections, which are well-written and substantively correct, although minor errors or shortcomings may be present. The overall structure is clear and coherent.

**Description Of Results:**

4: High Quality – The results are described in detail and supported by usage examples or evaluations. The description is reliable but may lack full depth of analysis.

**Feedback On Consistency:**

Consistency of the Project Description
The project description for "Project of Projects" (PoP) appears to be mostly consistent and logical.
Strengths:
* Clear Problem Statement: The introduction outlines the challenges of managing Team Projects (ZPI) and the need for a platform like PoP.
* Logical Flow: The report progresses logically, discussing existing solutions (see areas for improvement below), technology choices, results with key features, and future directions.
* Cohesive Argument: The text consistently emphasizes how PoP addresses the identified problems.
* Technical Details: The report signals the technology stack and implementation choices to demonstrate understanding. I assume that the limited length of the report influenced the depth of technical details provided.
Areas for Improvement:
* Related work not really related to what the authors aimed to build. The authors aimed to build “a tailored solution designed specifically to address the unique needs of our faculty in managing student [software] projects”. It is hard to treat Google Scholar (or other considered solutions) as related work to what the authors aimed to build.
* Minor Inconsistencies, e.g., the conclusion mentions "business perspective" which wasn't explicitly discussed earlier.
Overall, the report provides a brief but mostly easy-to-grasp picture of PoP's functionalities and its value proposition.

**Potential For Development:**

The technical report indicates possibilities for further work (Mobile App, Polish Version, Offline Access).

**Project Nature Evaluation:**

DISCLAIMER: I was not involved in the ZPI course this year so my comments may not reflect what was expected from the students during the course, and what was expected to be included in the submitted technical report. Also, even though the assessed artifact looks like a scientific paper, I assume that my role is not to assess scientific contributions if any (which is typically the task of article reviewers), but rather technical ones presented in the form of a final report from a ZPI software project.

The "Project of Projects" (PoP) developed by BSc students exhibits characteristics of engineering work.
Level of Utility:
* Addresses a Real-World Problem: PoP directly solves the issue of inefficient and manual management of Team Projects (ZPI) in an academic setting.
* Improves Efficiency: By automating workflows and providing a centralized platform, PoP aims to reduce the time and effort required for project management.
* Enhances Collaboration: The platform facilitates collaboration among students, supervisors, reviewers, and event chairs, leading to improved project outcomes.
Application of Technical Methods:
* Software Engineering: The project involves the design and development of a software application, but testing is only mentioned once and not described at all. Software development methodology is not discussed in detail.
* Database Design: The use of a relational database (PostgreSQL) to store and manage project data demonstrates the application of database design principles.
* Web Development: The frontend and backend development aspects involve using technologies like Svelte, Tailwind CSS, and Spring Boot, which are essential for building web applications.
Technological Solutions:
* Authentication and Authorization: The usage of JWT and OAuth protocols shows that the authors think about security.
* Workflow Management: The platform incorporates defined workflows for project submission, evaluation, and feedback.
* Data Analysis and Visualization: The statistics dashboard provides insights into project data, leveraging data analysis and visualization techniques.

**Technical Language Precision:**

4: High Quality – The language is appropriate for a technical report. Terminology is used correctly, and statements are precise, with only minor shortcomings that do not affect the overall clarity.

---

### Official Review · Reviewer_DZag · 2024-12-09
**Project of Projects**

**Confidence:** 2
**Significance Of Results:** 2
**Overall Quality:** 2

**Compliance With Template:**

3: Average Quality – The article includes most of the required sections, but some may be incomplete, written in a general or unclear manner. The content is correct but requires further refinement.

**Description Of Results:**

2: Low Quality – The results are described very superficially and in a general manner. Essential details, usage examples, or evaluations are missing.

**Feedback On Consistency:**

The comparison to google scholar or research gate is strange. They are not meant for project management, of management of common papers. Google docs would even be a better one. Or comparison to scientific journal submission systems would make more sense. Also MS Teams has functions, and Moodle for project management. These should be compared.

**Potential For Development:**

Yes.

**Project Nature Evaluation:**

To be honest, it was difficult to understand how such a project management app would function. Maybe due to the lack of technical knowledge from my side. But as an editor of a few scientific journals, and being involved in different scientific projects and student project, I do not understand the functioning of the app.

**Technical Language Precision:**

3: Average Quality – The language is mostly appropriate but may contain minor terminological or stylistic errors. Some statements might lack precision or require improvement for better readability.

---

### Decision · Program_Chairs · 2024-12-10

Accept (Poster)